# Enhancing NO Uptake in Metal-Organic Frameworks via Linker Functionalization. A Multi-Scale Theoretical Study

**Charalampos G. Livas** [1] , **Emmanuel Tylianakis** [1,2] **and George E. Froudakis** [1,*]

1    Department of Chemistry, University of Crete, Voutes Campus, GR-71003 Heraklion, Crete, Greece
2    Department of Materials Science and Technology, University of Crete, Voutes Campus, GR-71003 Heraklion, Crete, Greece
*    Correspondence: frudakis@uoc.gr

**Abstract:** In the present work, ab initio calculations and Monte Carlo simulations were combined to investigate the effect of linker functionalization on nitric oxide (NO)'s storage ability of metal–organic frameworks (MOFs). The binding energy (BE) of nitric oxide with a set of forty-two strategically selected, functionalized benzenes was investigated using Density Functional Theory calculations at the RI-DSD-BLYP/def2-TZVPP level of theory. It was found that most of the functional groups (FGs) increased the interaction strength compared to benzene. Phenyl hydrogen sulfate ($-OSO_3H$) was the most promising among the set of ligands, with an enhancement of 150%. The organic linker of IRMOF-8 was modified with the three top-performing functional groups ($-OSO_3H$, $-OPO_3H_2$, $-SO_3H$). Their ability for NO adsorption was investigated using Grand Canonical Monte Carlo (GCMC) simulations at an ambient temperature and a wide pressure range. The results showed great enhancement in NO uptake constituting the above-mentioned FGs, suggesting them to be promising modification candidates in a plethora of porous materials.

**Keywords:** nitric oxide storage; metal–organic frameworks (MOFs); multi-scale approach

## 1. Introduction

Nitric oxide (NO) or nitrogen monoxide is a colorless gas and one of the principal oxides of nitrogen ($NO_x$). The latter is a family of highly reactive gases that are largely released into the atmosphere as a result of the combustion of fossil fuels [1]. The majority (43%) of NO emissions originate from road transport and are also the dominant component (95%) of all gaseous $NO_x$ effluents [2]. Consequently, it is the focus of $NO_x$ removal efforts. Both the Environmental Protection Agency (EPA) of the United States of America and the European Environment Agency (EEA) regard NO as an important air pollutant among other gases such as $NH_3$ and $SO_2$ [3]. Its toxicity influences the environment and human health [4]. Air pollution and chronic exposure to hazardous gases can cause respiratory and cardiovascular diseases such as lung cancer and heart failure. Almost all of the global population (99%) are living in areas with poor air quality, a fact that leads to more than four million deaths every year, according to the World Health Organization (WHO) [5]. There are estimates that the global economic cost of tackling air pollution and its health effects has increased to USD 8.1 trillion which is equivalent to 6.1 percent of global GDP [6]. Moreover, $NO_x$ can react with water, oxygen, and other chemicals to produce acid rain [7]. It has negative biological consequences, especially in aquatic areas, and with dry deposition, it causes metal corrosion and can negatively impact buildings such as monuments [8].

Among the solutions being investigated to reduce NO emissions into the atmosphere is the discovery of new materials with high adsorption capacity for NO. Metal–organic frameworks (MOFs) have aroused a lot of attention as promising gas-adsorption materials. MOFs are organic–inorganic hybrid materials consisting of metal ions or clusters joined by organic linkers, as discovered by O. Yaghi in 1995 [9]. Most MOFs have an open porous

framework, which gives them essential features, including low density, large surface area, and high porosity. These characteristics result in the adsorption of gases in greater quantities and at lower pressures. MOF characteristics may be fine-tuned for each application because of the diversity and tunability of their structure. Gas storage, ion exchange, molecular separation, and heterogeneous catalysis are possible uses [10–15].

Important techniques have been used to improve the nitric oxide storage capacity of MOFs. To modify the nature of the pore surface and increase the interaction between NO and the framework, the organic linkers can be functionalized, and the metal nodes serve as binding sites for the unitary structure. Aside from the benefits outlined above, certain MOFs feature open-metal sites (OMSs), which might be used wisely for specific applications [16]. Xiao et al. conducted experimental measurements for NO adsorption on HKUST-1. The latter gave an adsorption capacity of 9.0 mmol·g$^{-1}$ at 196 K and 1 bar. At the same pressure and at 298 K the adsorption capacity was ~3 mmol·g$^{-1}$ [17]. In addition, McKinlay et al., synthesized and tested Fe-based MIL-88A, MIL-88B, MIL-88B-NO$_2$, and MIL-88B-2OH for NO adsorption capacity. MIL-88A performed the best, with an adsorption capacity of 2.5 mmol·g$^{-1}$ [18].

The aim of this study is to investigate and propose new MOF materials with improved nitric oxide adsorption capabilities. Following this pathway, IRMOF-8 was incorporated with strategically selected functional groups (FGs) in order to enhance the interaction between the gas and the material, thus increasing the adsorption capacity for nitric oxide. First, we calculate the binding energy between functionalized benzene rings and the NO molecule, and then, the top-performing functional groups are introduced on the phenyl ring of the organic linker of IRMOF-8 to investigate the impact on the NO adsorption profile.

Our group has applied the aforementioned strategy in several studies that focus on different gases, and the results are indicative of the effectiveness of linker functionalization [19–23]. For instance, Raptis et al. [20] showed that the introduction of numerous FGs increases the binding energy (BE) between the functionalized benzene ring and nitrogen dioxide by 170% compared to the unfunctionalized one. This increase leads to a 16-fold enhancement in gravimetric adsorption (mmol·g$^{-1}$) at 1.2 bar and 298 K. In addition, the work of Frysali et al. [21] revealed that the presence of a sulfate anion in the phenyl ring reinforces the interaction between the latter and the carbon dioxide ($-22.6$ kJ/mol) almost 2-fold.

Considering that aromatic rings are the building blocks of many MOFs' organic linkers, we investigated the interaction of a NO gas molecule with a series of 42 carefully selected substituted benzene rings. The generation of electron redistribution plots and electrostatic potential maps of the dimers and of the functionalized benzenes, respectively, was required to shed light on the nature of the interactions. Finally, in order to assess the effect of linker functionalization on the nitric oxide uptake profile, we conducted Grand Canonical Monte Carlo (GCMC) simulations for IRMOF-8 modified with the three top-performing FGs. The total gravimetric and volumetric uptake isotherms were obtained at 298 K and for a wide pressure range.

## 2. Methodology

Full geometry optimizations were employed using the ORCA program package [24,25] at the DSD-BLYP/def2-TZVPP level of theory [26–28] to investigate the binding strength between NO and a large set of 42 functionalized benzene molecules. Resolution of identity (RI) approximation was used along with the auxiliary basis set to speed up the calculation time [29]. The Basis Set Superposition Error (BSSE) was accounted for and corrected using the Counterpoise method as proposed by Boys and Bernardi [30].

In order to obtain a better understanding of the interactions between the dimers, we constructed electrostatic potential maps of the functionalized benzene monomers and electron density redistribution plots of the dimers. The visualization was carried out using the gOpenMol graphics program [31,32].

To shed light on the effect of linker functionalization on the adsorption capacities of NO in MOFs, IRMOF-8 was chosen and was modified by the three functional groups that

showed the greatest interaction with NO. The parent and modified organic linkers can be seen in Figure 1. Monte Carlo simulations were employed in the Grand Canonical ensemble at 298 K for a pressure range up to 100 bar using the RASPA software package [33].

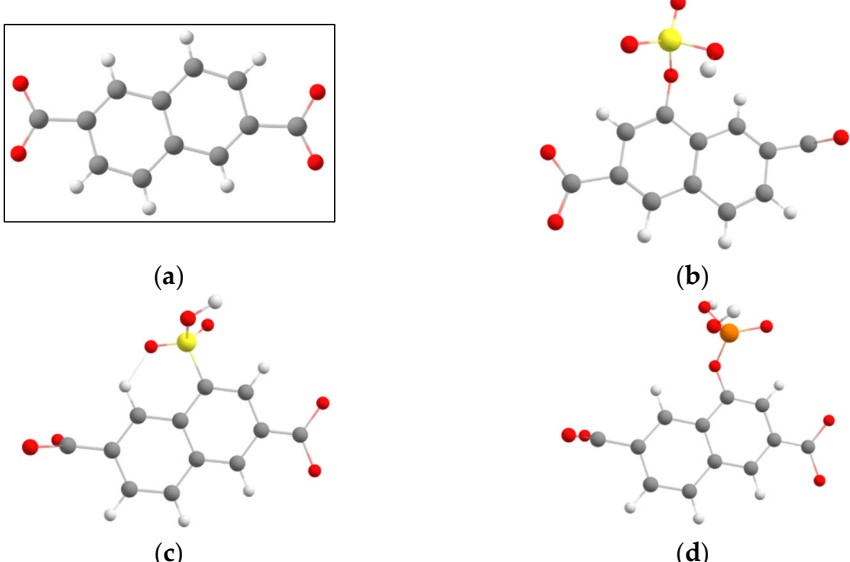

**Figure 1.** The original and functionalized linkers of the IRMOF-8 structures studied using GCMC simulations: (**a**) the parent IRMOF-8 linker; the (**b**) –OSO$_3$H, (**c**) –SO$_3$H, and (**d**) –OPO$_3$H$_2$ functionalized linkers. Carbon, hydrogen, oxygen, sulfur, and phosphorus atoms are depicted as gray, white, red, yellow, and orange spheres, respectively.

For the description of host–sorbate interactions, the Lennard-Jones [34] + Coulomb potentials were used. Equation (1) represents the aforementioned potential.

$$V_{ij} = 4\varepsilon_{ij}\left[\left(\frac{\sigma_{ij}}{r_{ij}}\right)^{12} - \left(\frac{\sigma_{ij}}{r_{ij}}\right)^{6}\right] + \frac{q_i q_j}{4\pi\varepsilon_0 r_{ij}} \tag{1}$$

Here, $\varepsilon_0$ is the vacuum permittivity constant, $q_i$ and $q_j$ are the corresponding partial charges for atoms $i$ and $j$, $r_{ij}$ is the interatomic distance between interacting atoms $i$ and $j$, and $e_{ij}$ and $\sigma_{ij}$ are the LJ potential well depth and the repulsion distance between atoms $i$ and $j$, respectively.

Nitric oxide was treated using the COMPASS model [35], and during the simulations, the bond length was 1.152 Å and was not allowed to vary. For the van der Waals interactions, the potential parameters used for the nitrogen and oxygen atom were $\varepsilon/k_b$ = 79.5 K, $\sigma$ = 3.0 Å, and $\varepsilon/k_b$ = 96.9 K, $\sigma$ = 2.8 Å, respectively. The corresponding parameters for each MOF framework were obtained via the UFF force field [36]. Lorentz–Berthelot mixing rules [37] were employed to describe the interactions between the MOF framework and the gas molecule. To better describe the intermolecular interactions between NO molecules and the functional group atoms, we fitted the classical potential parameters to the quantum chemical data. More information about the fitting procedure can be found in the Supplementary Material. The cutoff value for the LJ potential was set to 12.8 Å. Electrostatic interactions were also considered. Point charges of 0.0288e and −0.0288e were appointed to the nitrogen and oxygen atoms, respectively, for the electrostatic interactions. The partial charges of the MOF atoms were calculated as implemented in ORCA using the CHELPG method [38]. The resulting excess charges were split in order to keep the molecular structure in the periodic box neutral.

### 3. Results and Discussion

In this study, the NO interaction with 42 functionalized benzene molecules was theoretically investigated. In Figures 2 and 3, the geometries and the corresponding binding energies of the most promising systems can be seen. A detailed representation of the results for the remaining systems under study can be found in Figures S1 and S2 of the supporting information. The energetically most favorable structures, seen in Figure 3, can be divided in two categories: NO located above the benzene ring (Group 1) or towards the FG (Group 2). The binding energies calculated at the RI-DSD-BLYP/def2-TZVPP level, reported in Figure 2, range from 10.1 to 19.3 kJ/mol for NO–$C_6H_5$OOH and NO–$C_6H_5$OSO$_3$H, respectively. The weakest interaction, as seen in Figure S1, was with $C_6H_5$CN, with a binding energy of 6.6 kJ/mol.

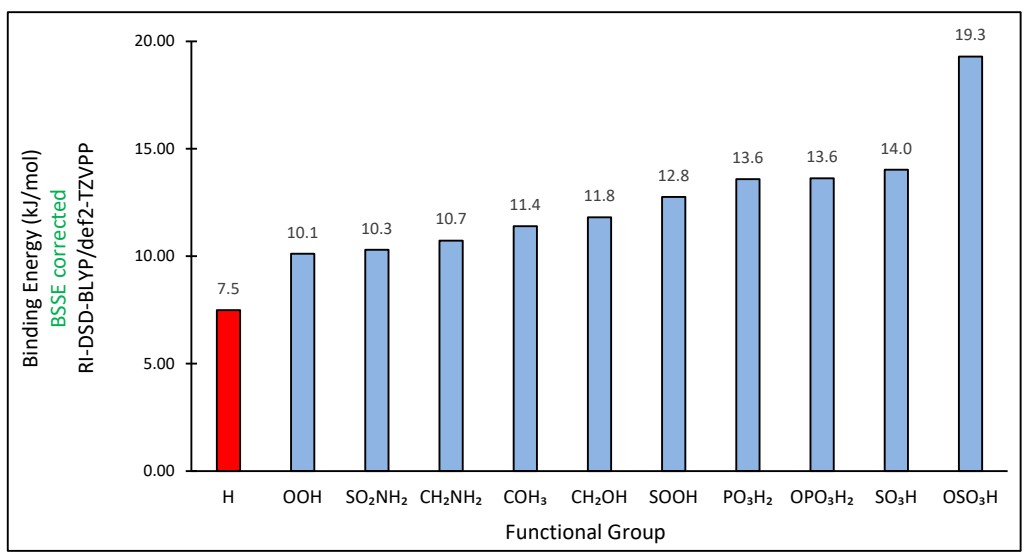

**Figure 2.** The 10 substituted benzenes (blue) with the highest NO binding energy and the unsubstituted benzene (red) for comparison, calculated at the RI-DSD-BLYP/def2-TZVPP level of theory and corrected for BSSE.

The functional group that showed the highest interaction energy was phenyl hydrogen sulfate with a corresponding binding energy of 19.4 kJ/mol, i.e., almost 3 times stronger binding than that of the unfunctionalized benzene ring (7.5 kJ/mol). As seen in Figure 3, RI-DSD-BLYP/def2-TZVPP optimized geometries for the 10 most energetically favorable dimers the NO molecule is above the phenyl ring, so it belongs to group 1 and is close to the FG. The nitrogen atom of the NO molecule is coordinated towards the hydrogen atom of the –OSO$_3$H functional group at a distance of 2.04 Å, which is the smallest among all the systems under study.

The functional groups that constitute the top-three-performing FGs, along with OSO$_3$H, are –SO$_3$H and –OPO$_3$H$_2$, which have binding energies of 14.0 kJ/mol and 13.6 kJ/mol, respectively. In contrast with the best one, –OSO$_3$H, the NO molecule in the aforementioned systems, is away from the benzene ring and is oriented next to the functional group, thus falling within group 2. We can see that the nitrogen atom of the NO molecule is close to the hydrogen atom of each of the FGs. Specifically, the corresponding distances are 2.10 Å for –SO$_3$H and 2.13 Å for –OPO$_3$H$_2$. These distances are the second and third smallest ones observed among all the dimers, as seen in Figure 4. We can also notice that the distance between the aforementioned N and H atoms increases as the binding energy decreases in an almost linear correlation.

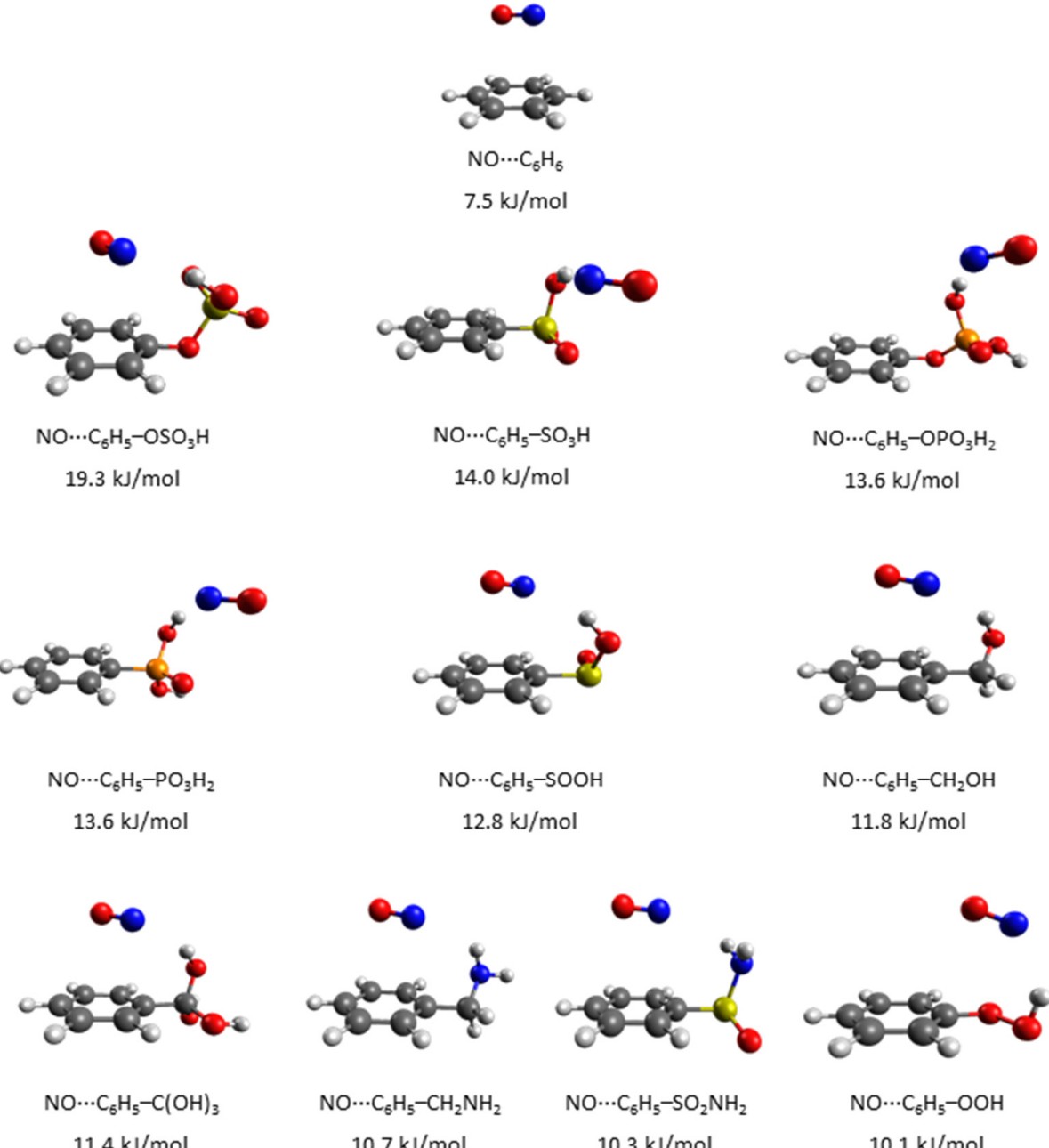

**Figure 3.** RI-DSD-BLYP/def2-TZVPP optimized geometries for the 10 most energetically favorable dimers.

Figure S1 shows the optimized geometries for all the systems under study. We can observe that nitric oxide orients itself above the benzene ring in parallel mode in most cases. It is also slightly displaced towards the FG (Group 1). This suggests that, in addition to the predicted interaction between the dipole moment of NO and the π system of the benzene monomer, there is an extra interaction between NO and the atoms of the polar groups. There are cases where the NO molecule is placed next to the functional group and away from the benzene ring (Group 2). In these dimers, the NO molecule is closer to the FG compared with the cases where it is above the ring, as seen by comparing Figure 5a,b. Thus, NO is taking advantage of the additional interaction between the nitrogen atom of NO and the hydrogen atom of the FG.

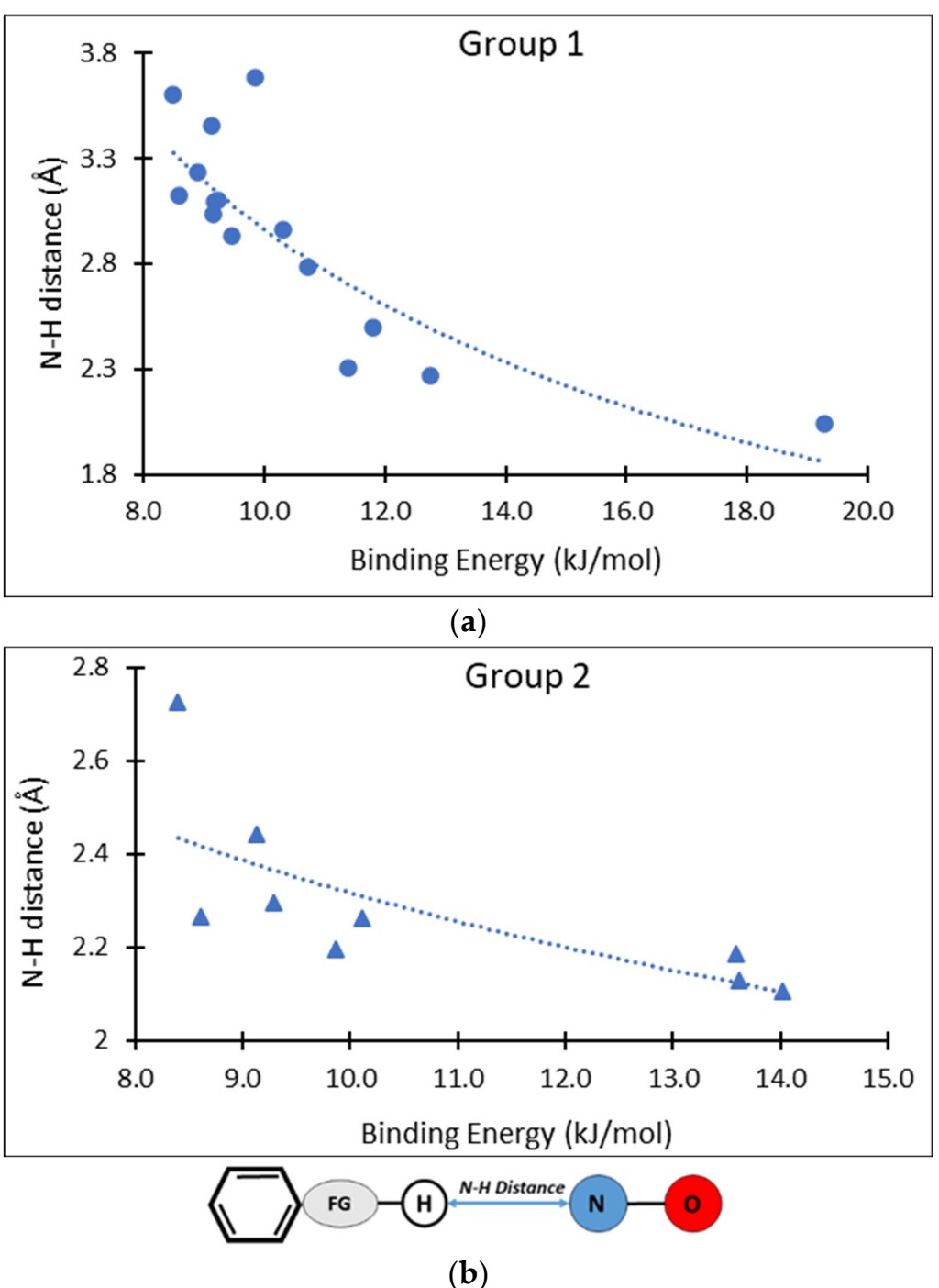

**Figure 4.** Plot of the distance between the nitrogen atom of the NO molecule and the hydrogen atom of the functional group as a function of the binding energy for (**a**) Group 1 and (**b**) Group 2.

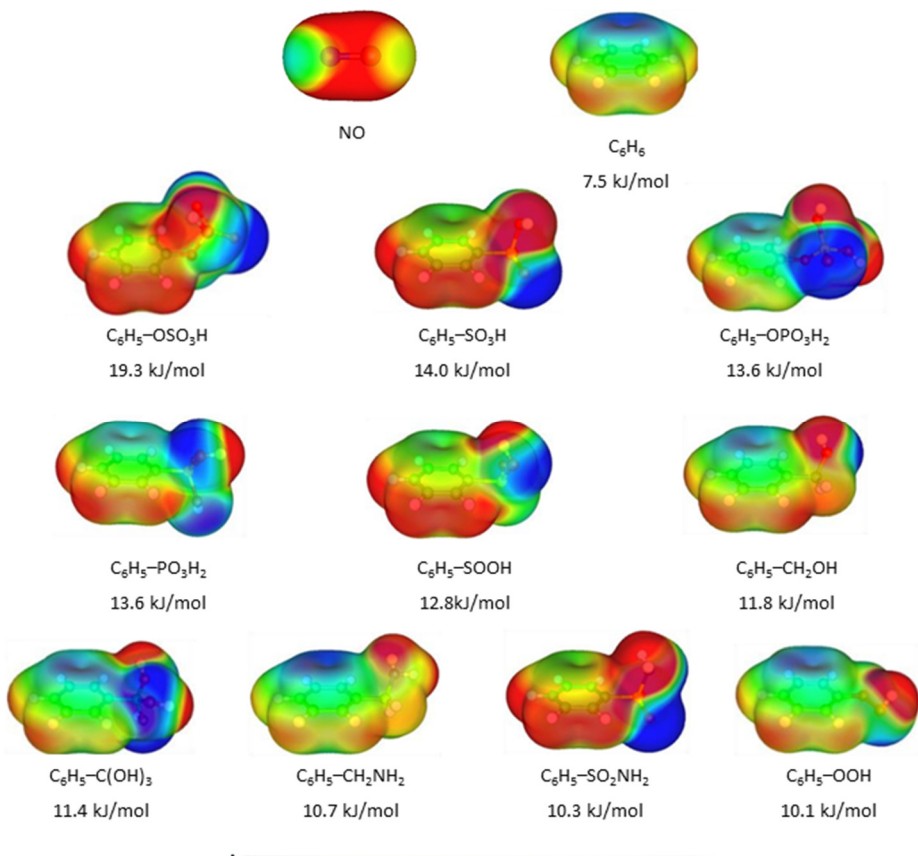

**Figure 5.** The electrostatic potential maps for the 10 most energetically favorable dimers. Calculated using ORCA 4.2 and visualized using gOpenMol [27,28]. The varying intensities range from −0.03 to +0.03 Hartree·e$^{-1}$. Red: electron-poor regions—high-potential value; blue: electron-rich regions—low-potential.

Aiming to shed light on the nature of interactions, electrostatic potential maps were produced for all the functionalized benzene rings studied in this work. By doing so, we are capable of visualizing the charge distribution of the monomers and explaining the optimized geometries. In Figure 5, the electrostatic potential maps of the top-10-performing substituted benzene rings can be seen, and they can be seen in Figure S3 for all the benzene monomers under study. In both, we can see the corresponding map for the NO molecule, which has a high-potential region in the middle and two low-potential regions at both ends of the molecule. This, combined with the observation that the middle of all the benzene ring molecules have a low-potential value, plays an important role in the preference of the gas molecule being located above the benzene ring. In addition, the nitrogen atom of the NO molecule has a higher density of electrons than the oxygen atom. This explains the orientation of the N atom towards the hydrogen atom of the functional group which, as seen below, has a low electron density.

The next step in our study is to validate the functionalization's effect in increasing NO adsorption in MOFs. To this end, IRMOF-8 was selected and modified with the top-three-performing functional groups. As seen in Figure 1, the organic linker of IRMOF-8 is 2,6-naphthalene dicarboxylate and it was functionalized. The gravimetric and volumetric uptake isotherms, which are shown in Figures 6 and 7, were obtained for pressure up to 100 bar and at a temperature of 298 K using Grand Canonical Monte Carlo simulations.

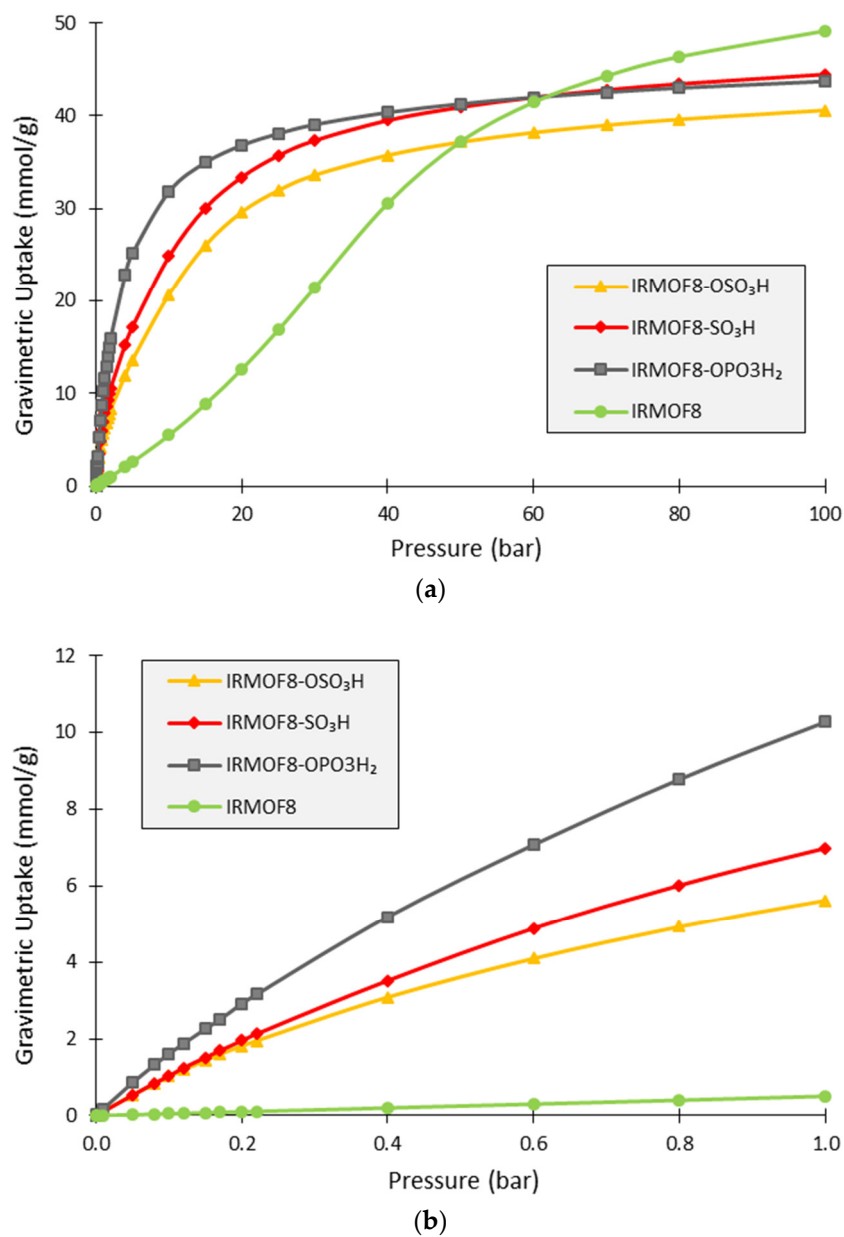

**Figure 6.** Absolute Gravimetric isotherms for IRMOF8 and IRMOF8-n (n: –OSO$_3$H, –SO$_3$H, and –OPO$_3$H$_2$,) at T = 298 K and pressure range up to (**a**) 100 bar and (**b**) 0.10 bar.

In terms of the gravimetric uptake, we see a great enhancement, especially in the region of pressure lower than 40 bar. At ambient pressure, i.e., 1 bar, we see a 20-fold increase for the best-performing one, IRMOF8-OPO$_3$H$_2$. Concerning the uptake profile for pressure above 40 bar, we see that the parent material behaves better than the modified ones. This is explained by the fact that the gravimetric uptake is the amount of gas adsorbed divided by the weight of the system. By adding the functional groups to the framework, we increase the denominator. At larger pressures where the key factor is the pressure itself and not the binding energy between the framework and the gas molecule, we see similar quantities of gas adsorbed. So, the nominator for the different materials is similar but the denominator is smaller for the parent material, i.e., IRMOF-8.

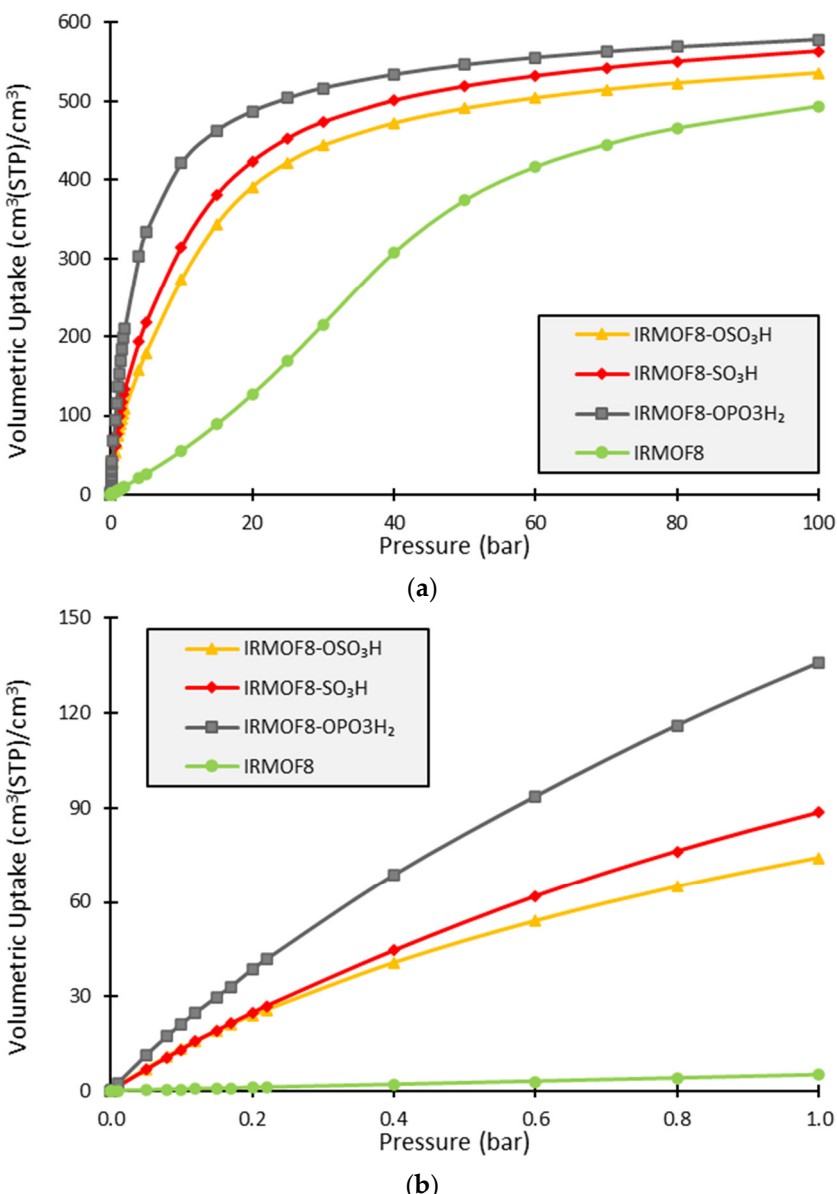

**Figure 7.** Absolute volumetric isotherms for IRMOF8 and IRMOF8-n (n: –OSO$_3$H, –SO$_3$H, and –OPO$_3$H$_2$,) at T = 298 K and pressure ranges up to (**a**) 100 bar (**b**) 0.10 bar.

Concerning the volumetric uptake, we produce analogous results, as demonstrated in Figure 7. At the low loading limit, the performance of the modified materials is improved and at a pressure above 40 bar, the difference between the modified IRMOF-8 materials and the parent one is decreasing. The parent material cannot surpass the volumetric uptake of the modified ones because the denominator of this indicator, i.e., cm$^3$(STP)/cm$^3$ is the same for all systems. By comparing the latter results with the highest nitric oxide uptake under these thermodynamic conditions found in the literature [16], one can conclude that the functionalization of MOFs is an important technique for the improvement of gas uptake. More specifically, Xiao et al. [17] found a 3 mmol/g storage capacity for HKUST-1 at 298K and 1 bar, which constitutes one third of the adsorption of IRMOF8-OPO$_3$H$_2$ in the same thermodynamic conditions.

We notice that the IRMOF-8 functionalized with –OPO$_3$H$_2$ has greater adsorption capacity than the ones modified with –OSO$_3$H and –SO$_3$H. This is in contrast with the binding energies of the three dimers of C$_6$H$_5$-X (X = –OSO$_3$H, –SO$_3$H, and –OPO$_3$H$_2$) as the order is exactly the opposite. We can attribute this to the fact that the binding energy of

the global minimum is an important indicator of the adsorption capabilities of the material, but it is not the only one that plays a crucial role. The different binding sites and the corresponding binding energies have a strong impact on the adsorption capacity. Figure 8 is a schematic representation of the different local minima obtained after analyzing the data of the DFT study, with a more detailed depiction shown in Table S1 and Figure S7. We can see that $C_6H_5$-$OPO_3H_2$ has, on average, greater binding energies than $C_6H_5$-$OSO_3H$ and $C_6H_5$-$SO_3H$. So, in addition to $C_6H_5$-$OPO_3H_2$ having lower binding energy for the first NO molecule, it has better binding sites for the other approaching NO molecules. This translates to greater adsorption capacities in GCMC simulations. The same applies to $C_6H_5$-$SO_3H$ and $C_6H_5$-$OSO_3H$, and that is why we observe lower adsorption for the latter.

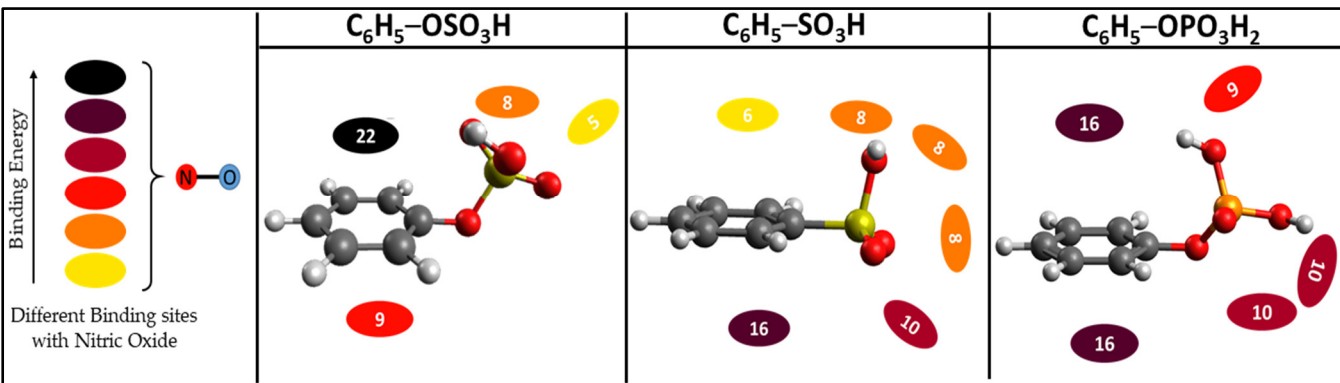

**Figure 8.** Schematic representation of the different binding sites of the three top-performing functional groups (–$OSO_3H$, –$SO_3H$, –$OPO_3H_2$). The numbers in the oval shapes are the corresponding binding energies in kJ/mol calculated at the RI-DSD-BLYP/def2-TZVPP level of theory (without BSSE correction).

## 4. Conclusions

In this study, 42 functionalized benzene rings were examined by means of QM calculations concerning their interaction strength with the NO molecule. Almost all the functional groups show enhanced interaction when compared to the parent ring with the –$OSO_3H$ group, showing a binding energy of −19.3 kJ/mol. The nature of the interactions was further analyzed using electrostatic potential maps for the modified rings and redistribution electron density maps for the optimized dimers with NO molecules. The impact of functionalizing the benzene ring with the best-performing functional groups, i.e., –$OSO_3H$, –$SO_3H$, and –$OPO_3H_2$, was examined by performing Grand Canonical Monte Carlo simulations on the accordingly modified IRMOF-8 structures. The simulations were conducted at 298 K and up to 100 bar. All the functionalized materials show increased uptake under ambient conditions in both volumetric and gravimetric terms. Overall, this study establishes a systematic and transferrable database that may be utilized for additional experimental and computational investigations on the augmentation of the NO adsorption capabilities of porous materials.

**Supplementary Materials:** The following supporting information can be downloaded at: https://www.mdpi.com/article/10.3390/chemistry4040086/s1, Figure S1: Binding energies (kJ/mol) of the NO···$C_6H_5$-X dimers, calculated at the DSD-BLYP/def2-TZVPP level of theory. Basis Set Superposition Error (BSSE) was taken into consideration by the full counterpoise method; Figure S2: More energy-favorable configurations for the systems in this work; Figure S3: Electrostatic potential maps of all substituted monomers $C_6H_5$-X. Calculated using the DSD-BLYP/def2-TZVPP method with ORCA 4.2 and visualized with gOpenMol. The varying intensities are ranging from −0.03 to +0.03 Hartree·$e^{-1}$. Red: Electron-poor regions-high potential value, Blue: electron-rich regions-low potential; Figure S4: Fitting of the ($\varepsilon/k_b$, $\sigma$) parameters of the UFF potential on the QM data obtained from the DFT scan of NO over benzene; Figure S5: Fitting of the ($\varepsilon/k_b$, $\sigma$) parameters of the UFF potential on the QM data obtained from the ab-DFT scan of NO over (a) $C_6H_5$-$OSO_3H$ (b) $C_6H_5$-$SO_3H$ (c) $C_6H_5$- $OPO_3H_2$; Figure S6: Gravimetric (a) and Volumetric (b) Nitric Oxide uptake

isotherms for IRMOF-8 and IRMOF-8-n (n: –OSO$_3$H, –SO$_3$H, –OPO$_3$H$_2$); Figure S7: The geometries of the different local minima (RI-DSD-BLYP/def2-TZVPP); Table S1: The binding energies of the different local minima calculated at the RI-DSD-BLYP/def2-TZVPP level of theory (without BSSE correction); [24,25,30–32,36,37].

**Author Contributions:** Conceptualization, C.G.L. and G.E.F.; methodology, C.G.L., E.T. and G.E.F.; software, C.G.L. and E.T.; validation, C.G.L. and E.T.; formal analysis, C.G.L.; investigation, C.G.L.; resources, G.E.F.; data curation, C.G.L.; writing—original draft preparation, C.G.L.; writing—review and editing, C.G.L. and G.E.F.; visualization, C.G.L.; supervision, G.E.F. All authors have read and agreed to the published version of the manuscript.

**Funding:** European Regional Development Fund of the European Union and Greek national funds through the Operational Program Competitiveness, Entrepreneurship, and Innovation, under the call RESEARCH—CREATE—INNOVATE: T2EΔK-01976.

**Institutional Review Board Statement:** Not applicable.

**Informed Consent Statement:** Not applicable.

**Data Availability Statement:** The data presented in this study are available in Supplementary Materials.

**Conflicts of Interest:** The authors declare no conflict of interest.

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
