# Peer review of "Enhancing NO Uptake in Metal-Organic Frameworks via Linker Functionalization. A Multi-Scale Theoretical Study"

_chemistry, doi:10.3390/chemistry4040086_

Round 1

Reviewer 1 Report

The work is a computational study of the NO uptake of Metal-Organic Frameworks upon judicial modification of the organic linkers.
The computational methods employed are on two levels of theory; the first level is Density Functional Theory (DFT) and the second is a combination of grand canonical Monte Carlo simulations with suitably parameterized classical force fields.
The authors have chosen a double hybrid function which is computationally very expensive and is expected to provide highly accurate results. The basis set is more that suitable to further provide highly accurate results. As a note, I would have expected the authors to include diffuse functions in the basis set but the lack of these is not expected to produce noteworthy errors.

The authors proceed to parameterize classical force field to include long range interactions based on results they obtain from the higher level of theory. The interactions are of Lennard-Jones + Coulomb potentials and are very well accepted in such works.

Multiple and judiciously chosen configurations were examined. The mapping of the relative configurations is thorough.

The discussion of the results are appropriate for both the ab initio results and the grand canonical Monte Carlo simuations. The discussion is also comparative. A considerably large pressure range has been examined.

The present work is contemporary and of high interest to a rather broad audience. The methodology followed is solid and the discussion of results extensive. The use of English is clear througout the manuscript.

I suggest that the work be published.

1) There is a point that may lead to misinterpretation and should be clarified. \epsilon here is given in units of K, however \episilon should be given in units of energy. Alternatively, \epsilon / k_B can be provided in units of Κ. The authors should fix this. The absolute values themselves seem reasonable.

2) The authors may want to consider providing some details on the computational cell setup. This is not mandatory or prerequisite for my verdict, only a suggestion. An image of the cell would also be well received.

The reference numbering has produced errors. It was quite a task to identify the citations from the references. That being said, the references are all suitable and up to date.

Reviewer 2 Report

The authors did work on functionalization of IRMOF-8 and studied uptake of NO in functionalized IRMOF-8 using DFT. I think that this work is novel and significant to publish in this journal after a minor revision

1.    Ttile is too long, please shorten it

2.    There should be more discussion between the structure and catalytical performance. Besides, the performance is suggested to be compared with other literatures.

3.    Comparison should be given with already reported work how authors funtionalized MOF are better for NO uptake.

4.    The conclusion part should be rewritten to summarize the main work and suggested to be briefer and more succinct.

One concern from my side; now a day DFT study researcher did but without experimental work how can we justify work or result is it better or not?

Reviewer 3 Report

Detailed comments:

1.      The English of the text should be checked

2.      At Figure 2, where are indicated the level of theory and corrected for BSSE? In the figure is represented only BSSE corrected! Also, at axes, at scale, use the option “Tick Marks”. For formula at functional group, for number, use the option “subscript”

3.      At Figure 4, at axes, at scale, use the option “Tick Marks”

4.      Same Reference are not recognized by the system, as such the message always appears in the manuscript In Error! Reference source not found.  This problem must be fixed, so that the manuscript does not appear with this message but with the reference number

5.      For Figures 6 and 7, at legend, please correct the functional group, use the option “subscript”

6.      Comparison between the obtained results and measured in this study with other reported studies should be done and included for more clarity (indicate values not just number of reference).

7.      The References are very old. Some could be replaced with new article or re-edited books. The manuscript must contain the relevant information to be attractive for readers (researchers), because science has advanced, and the information indicated in the manuscript is no longer valid. This part should include observed information, noted in the last 10-12 years. 
